# Differential Expression of α-Enolase in Clinical Gastric Tissues and Cultured Normal/Cancer Cells in Response to *Helicobacter pylori* Infection and *cagA* Transfection

**DOI:** 10.3390/medicina58101453

**Published:** 2022-10-14

**Authors:** Feiyan Yu, Mengya He, Jian Li, Haiyan Wang, Shuaiyin Chen, Xiaojuan Zhang, Huijuan Zhang, Guangcai Duan, Rongguang Zhang

**Affiliations:** 1The First Affiliated Hospital, International School of Public Health and One Health, Hainan Medical University, Haikou 570102, China; 2Department of Epidemiology, College of Public Health, Zhengzhou University, Zhengzhou 450001, China; 3Henan Provincial People’s Hospital, Zhengzhou University, Zhengzhou 450003, China; 4School of Basic Medical Sciences, Henan University of Science and Technology, Luoyang 471000, China

**Keywords:** *Helicobacter pylori*, α-enolase, gastric cancer, cytotoxin associated gene A, precancer lesion

## Abstract

*Background and Objectives:* The role of α-enolase (ENO1) in *Helicobacter pylori*-related gastric lesions might be a critical factor in the pathogenesis, but remains undefined. *Materials and Methods:* This study investigated the differential expression of α-enolase in clinical gastric specimens and cultured normal/cancer cells in response to *H. pylori* (*cagA+*) infection and *cagA* transfection using qPCR, Western blots and histochemical methods. *Results:* A total of 172 gastric specimens were collected from 142 patients, the former comprising chronic superficial gastritis (CSG), precancerous diseases (PCDs, including atrophic gastritis, intestinal metaplasia and dysplasia) and gastric cancer (GC) cases. Among the CSG and PCD cases, the *H. pylori*-infected group had significantly elevated ENO1 mRNA levels compared with the uninfected group. In the GC cases, differential ENO1 expressions were detected between the cancer tissues and the paracancerous tissues. Notably, significant difference was first detected between the GC cell (AGS) and the normal cell (GES-1) as a response of ENO1 to *H. pylori* infection and *cagA* transfection. *Conclusions:* This report reveals that ENO1 expression is associated with *H. pylori* infection, *cagA* transfection, co-culture duration, multiplicity of infection, gastric normal/cancerous cell lines and cellular differentiation. The findings may be crucial bases for further ascertaining *H. pylori* pathogenic mechanism and formulating novel therapeutic and diagnostic strategies.

## 1. Introduction

Gastric cancer is one of the most common human carcinomas, with the second highest mortality rate among tumors [1]. *H. pylori* infection is thought to be the leading risk factor for gastric cancer [2]. Currently, it is known that *H. pylori* infection is closely related to a variety of gastrointestinal/extragastrointestinal diseases, including chronic superficial gastritis (CSG), chronic atrophic gastritis (CAG), intestinal metaplasia (IM), dysplasia, gastric cancer, liver cirrhosis, etc. [3,4,5]. However, host–microbe interactions behind these diseases (especially stomach cancer) remain poorly understood [3,4,5]. This issue has been hindering the development of preventive and therapeutic strategies for these diseases. 

The *H. pylori* cytotoxin-associated gene A (*cagA*) is a pathogenic virulence factor, positive in 60~70% of *H. pylori* strains [6,7]. The patients infected with *H. pylori cagA*+ strains are at higher risk for gastric cancer than the *cagA*- controls [8]. *H. pylori* can inject CagA protein into host cells via a type IV secretion system (T4SS), where CagA is phosphorylated and activates protein tyrosine phosphatase SHP-2 and the MAPK/MEK/ERK signaling pathway, thereby accelerating cellular proliferation, movement, diffusion and morphological changes [9,10]. CagA may have pathogenic effects by regulating cluster of differentiation (CD) antigens’ expression and metabolite pathways, e.g., glycolysis pathways [11,12]. The supposed roles of CagA in the pathogenesis need to be further evidenced.

α-enolase (ENO1) is a key glycolytic enzyme in catalyzing the conversion of phosphoglycerol to phosphoenol pyruvate, which is important for tumor proliferation, invasion, metastasis and adaption to hypoxic microenvironment [13]. One study reported that ENO1 affected gene transcription, cell apoptosis and differentiation, regulated expression of cancer protein C-myc through the Notch signaling pathways and thus probably contributed to tumor formation [13]. Few studies show links between *H. pylori* infection and ENO1 expression in cancer cells. It was observed that up-regulated ENO1 expression in gastric cancer cells co-cultivated with *H. pylori* [14]. Our group previously observed that the transfection of gastric cancer cells with *cagA* could cause high expression of ENO1 [10]. However, the effects of both *H. pylori* infection and *cagA* on ENO1 expression in normal and especially precarcinoma tissues of human stomachs remain unclear, which may be very important for further uncovering *H. pylori* pathogenesis.

This study investigates the differential expression of α-enolase in clinical gastric specimens and cultured normal/cancer cells in response to *H. pylori* infection and *cagA* transfection. It does so using clinical gastric specimens, co-culture of bacteria and gastric normal/cancerous cells and cellular transfection. This work is of significance for further revealing *H. pylori* pathogenic mechanism and for formulating novel diagnostic and therapeutic strategies targeted at ENO1 for gastric precancer lesions and cancer.

## 2. Materials and Methods

### 2.1. Clinical Specimen Collection and Detection

Collection of specimens: gastric mucosal tissue specimens and clinical data of CSG, CAG, IM and dysplasia cases were collected from patients inspected in the gastroscope room of Henan Provincial People’s Hospital from March to July 2018. Formalin-fixed gastric cancer tissues and the adjacent tissues were collected from the pathology department of this hospital from June to December 2018. 

Inclusion criteria: the patients included had never taken non-steroidal anti-inflammatory drugs, antibiotics or proton pump inhibitors for four weeks, had signed the informed consent of patients form and had been diagnosed with the diseases mentioned above by two physicians. The included cancer cases had not received any anticancer therapeutic treatments. The project was approved by Zhengzhou University Institutional Review Board and complies with the ARRIVE guidelines.

Extraction of total RNA and DNA from gastric tissues: a part of the CSG, CAG, IM and dysplasia specimens were kept in −80 °C for extraction of mRNA; the remaining parts were embedded in paraffin and sectioned using conventional methods. Total RNA of gastric tissues was extracted from the specimens using Trizol methods. DNA of gastric cancer and adjacent tissues was obtained from paraffin sections using the reported modified method [15] and QIAamp DNA Mini Kit (QIAGEN, German). In brief, five 5 μm-thick tissue slices were cut off from the paraffin-embedded gastric cancer and paracancerous tissues and put into a 1.5 mL centrifuge tube. Following addition of 1 mL xylene, the tube was vibrated violently for 10 s on a vortex oscillator and then centrifuged at room temperature at 16,000× *g* for 2 min. The supernatant was discarded and 1 mL anhydrous ethanol was added. After centrifugation for 2 min, the sediment was resuspended with 180 μL of ATL Buffer solution and then mixed with 20 μL proteinase K. The tube was placed at 56 °C for about 1 h until the tissue samples were completely dissolved. During this period the tube was turned upside-down 3~5 times. The mixture was incubated in 90 °C water for 1 h, then centrifuged for 2 min. The supernatant was moved to another centrifugal tube, to which was added 400 μL of mixture of Buffer AL and anhydrous ethanol (1:1 in volume). After transient centrifugation, the supernatant was transferred to the adsorption column and settled for 5 min. Post centrifugation at 12,000× *g* rpm for 1 min, the column was moved to a collection tube, then washed sequentially with 500 µL Buffer AW1 and 500 µL Buffer AW2. Finally, the DNA of the tissues was eluted from the adsorption column with 40 µL Buffer AE via centrifugation at 16,000× *g* for 3 min.

RNA reverse transcription: RNA reverse transcription was performed using PrimeScript™ RT reagent Kit with gDNA Eraser (Takara, China). The parameters of reverse PCR were 37 °C for 15 min, 85 °C for 5 s and 4 °C for 24 h.

PCR tests for *H. pylori* infection: *H. pylori* infection was determined by detection of *H. pylori* 16S rDNA in the gastric specimens. The PCR primers were 5′-TTTGTTAGAG AAGATAATGACGGTATCTAAC-3′ (sense) and 5′-CATAGGATTTCACACCTGACT GACTATC-3′ (anti-sense). The expected PCR product of *H. pylori*-positive specimens was 154 bp in length [16]. The cDNA from the CSG, CAG, IM and dysplasia specimens and DNA from the cancer and paracancerous specimens were used as the PCR templates. A case with either cancer or adjacent tissues positive for the 16S rDNA was diagnosed as *H. pylori*-positive.

Evaluation of ENO1 mRNA: ENO1 mRNA in gastric mucosal tissues was evaluated using quantitative PCR. The cDNA was used as the template. ENO1 primers and internal primers for testing for 18S ribosomal RNA used are shown in Table 1. Each sample was tested for three repeats using the TB Green Premix Ex Taq II (Tli RNaseH Plus) Kit (Takara, Dalian, China). 

Assessment of ENO1: ENO1 concentration in gastric tissues was assessed and subcellularly located via immunohistochemical staining. Rabbit anti-ENO1 antibody (Abcam, China) diluted at 1:200 was used as the primary antibody with the rabbit anti-actin antibody (Abcam, China) as the control and horse radish peroxidase-labeled goat anti-rabbit IgG antibody (Bioss, China) as the secondary antibody. The positive reaction was visualized using DAB, and the tissue section was re-stained with Harris’ haematoxylin (Servicebio, China). Photos of the stained tissues were taken under microscope, and the average optical density of the tissues was assessed and analyzed using the software ImageJ.

### 2.2. Co-Cultivation of Gastric Cells and H. pylori

*H. pylori* MEL-Hp27 (*cagA*+, *vacA*+) was isolated from a Zhengzhou patient with chronic atrophic gastritis. GES-1 (a normal gastric epithelial cell line) and AGS (a gastric cancer cell line) were supplied by Shanghai Cell Bank, Chinese Academy of Sciences (Shanghai, China) and Cell Room of Central South University (Changsha, China), respectively. *H. pylori* MEL-Hp27 was cultivated at 37 °C, 5% O_2_, 10% CO_2_ and 85% N_2_ using Brucella agar medium containing sheep blood (80 mL/L) as previously described [17]. The gastric cells were cultured at 37 °C, 5% CO_2_ with DMEM medium comprising fetal bovine serum (FBS, 100 mL/L).

AGS and GES-1 were respectively co-cultured with *H. pylori* for 24 h at multiplicity of infection (MOI) of 50:1, 100:1 and 200:1. Gastric cell cultures free from *H. pylori* were used as controls. Furthermore, *H. pylori* was co-cultured with GES-1 and AGS at MOI of 200:1 for 2 h, 6 h, 12 h and 24 h. The ENO1 mRNA in the cells was detected by qPCR.

### 2.3. Construction of CagA Expression Vector 

Genomic DNA of *H. pylori* MEL-Hp27 was extracted using DNA Extraction Kits (Lifefeng, China). *H. pylori cagA* gene was obtained by PCR from the genomic DNA using the primers as follows: P1, 5′-ACTAGTCCAGTGTGGTGGAATTCGCCACCATG ACTAACGAAAC-3′ (*EcoR* I) and P2, 5′-GTTTAAACGGGCCCTCTAGACTCGAGTT AAGATTTCTGGAAAC-3′ (*Xho* I). There is a Kozak sequence (GCCACC) in primer P1. 

The eukaryotic expression vector pcDNA3.1(+) was offered by Prof. Qiao Zhang and ligated to *cagA* using ClonExpress^®^ II One Step Cloning Kit (Vazyme, China), as instructed by the manufacturer. The ligation product was used to transform E. coli DH5α competent cells (Takara, China) using the heat shock method. The transformants were screened with ampicillin (100 µg/mL)-containing Luria Broth agar plates. The plasmid was extracted using plasmid extraction kits (Lifefeng, China), and sequenced by BGI Co. (Beijing, China). 

### 2.4. Cellular Transfection and Expression

The recombinant plasmid pcDNA-*cagA* was extracted using GoldHi EndoFree Plasmid Maxi Kits (CWBio, Taizhou, China), and introduced into AGS and GES-1, respectively, via liposome transfection using Lipofectamine^®^ 2000 Transfection Reagent (Invitrogen, Waltham, MA, USA). The plasmid pcDNA3.1(+) was used as the control. Total proteins and mRNA were extracted from the cultivated cells 72 h post transfection. The CagA protein expressed in the gastric cells was identified by Western blot using mouse anti-CagA antibody (Santa Cruz Biotechnology, CA, USA) and goat anti-mouse IgG antibody (Bioss, Beijing, China) as the primary and secondary antibodies, respectively. The positive reaction was visualized using ECL luminescence reagent (Absin, Shanghai, China). CagA expression level was determined using the software ImageJ.

For evaluation of ENO1 mRNA and *cagA* mRNA, cellular total RNA was extracted from the transfected AGS and GES-1 cells using the routine Trizol method. cDNA was obtained via reverse transcription, and both ENO1 and *cagA* mRNAs were assayed via qPCR. The ENO1 PCR primers used were as mentioned above, and those for *cagA* were 5′-ACGCTCTGTCTTCTGTGCTAATGG-3′ and 5′-AATCATGCCTAGCTTCAGGACCAC-3′. 

### 2.5. Statistical Analyses

The results were analyzed with the SPSS 21.0 software package. The measurement data are represented as mean ± standard deviation. Mann–Whitney rank sum U test was applied for non-normal data. T test and single-factor variance analysis were used for normal data. Comparisons of the groups were performed using Bonferroni method. Spearman rank correlation analysis was used in correlation analysis. The standard for the statistical tests was 0.05.

## 3. Results

### 3.1. Collection of Clinical Specimens

A total of 172 gastric mucosal tissue specimens were collected from 142 patients, including 73 cases of CSG, 5 cases of CAG, 29 cases of IM and 5 cases of dysplasia, 30 cases of gastric cancer and 30 adjacent tissues. The age, gender and *H. pylori* infection distribution of the CSG, precancerous diseases (PCDs, including CAG, IM and dysplasia) and gastric cancer patients is shown in Table 2.

### 3.2. Testing Results of the Gastric Specimens

As confirmed by immunohistological tests, ENO1 expression levels were significantly higher in the cancerous tissues of moderate/high differentiation grade than in those of low grade. ENO1 expression was enhanced in the cancerous tissues compared with the adjacent tissues (*p* < 0.001), and mainly located in the cellular membrane and cytoplasm (Figure 1). No statistical relation was found between ENO1 expression and *H. pylori* infection, age and lymphatic metastasis in the cancerous tissues (Table 3). Among the CSG and PCD cases, the *H. pylori*-infected group had significantly elevated ENO1 mRNA levels compared with the uninfected group (Table 4).

### 3.3. Co-Culture of Gastric Cells with H. pylori

Co-culture of the gastric cells with *H. pylori* for 24 h showed that the ENO1 expression in the AGS increased with the increase of MOI used, whereas the GES-1 cells lowered the ENO1 expression at a MOI of 50:1 (*p* < 0.001) in comparison to the uninfected control, and then enhanced the ENO1 expression accompanying the increase of MOI (Figure 2).

At MOI = 200:1, the ENO1 expression in the AGS cells was significantly down-regulated when co-culturing with *H. pylori* for 2 h or 6 h (*p* < 0.05), and up-regulated in co-culture for 12 h and 24 h (*p* < 0.05). Among the four set time points (*p* < 0.05), the GES-1 cells had elevated ENO1 expression only when co-culturing for 24 h (*p* < 0.05) (Figure 3).

### 3.4. Cellular Transfection and Impact on ENO1 Expression

The eukaryotic expression plasmid pcDNA-*cagA* was successfully constructed, as confirmed by PCR, restriction enzyme digestion and sequencing. The pcDNA-*cagA* was successfully transfected into GES-1 and AGS cells, and CagA was expressed, as confirmed by qPCR and Western blot (Figure 4). Compared with the control group, the ENO1 mRNA levels in both AGS and GES-1 cells were significantly enhanced after transfection (*p* < 0.05) (Figure 5). Furthermore, the ENO1 mRNA level in the cells was related to the *cagA* mRNA level (*p* < 0.05).

## 4. Discussion

Several methods have been established for the detection of *H. pylori* colonization in the stomach, including histopathological staining, PCR and bacteria culture and urea breath test [18]. To confirm *H. pylori* infection, the present study adopted PCR method for a high sensitivity [18]. In addition, as reported, GAPDH, ACTB, RP II and 18S ribosomal RNA (rRNA) are commonly used as the internal control genes in qPCR tests [19]. While the GAPDH, ACTB and RP II expressions may change with the severity of gastric pathological lesions, the 18S ribosomal RNA can be kept at a relatively stable level [19]. Therefore, the present study adopted the 18S rRNA gene as the internal control. 

Using the clinical specimens, the present study observed elevated ENO1 levels in the gastric cancerous tissues in comparison with the paracancerous tissues. This finding is identical to a previous report [14]. Similarly, neuron-specific enolase, another enolase subtype homogenous to ENO-1, was higher in sera of gastric adenocarcinoma cases than in those of healthy individuals [20]. These observations indicate up-regulated glycolysis in gastric cancer tissues. In addition, significant difference was also detected in ENO1 expression levels between the cancer tissues of low differentiation and those of high/moderate differentiation, suggesting the correlation of ENO1 with cellular differentiation. 

Unexpectedly, no significantly differential expression of ENO1 was detected in the cancerous tissues between the *H. pylori*-positive group and the negative group of the gastric carcinoma cases. In the cultured gastric cancer cells the *H. pylori*-infected group had significantly enhanced ENO1 expression compared with the uninfected control group. The in vitro observations are divergent from the in vivo findings in the cellular experiments, suggesting the existence of an unknown mechanism. The microecological difference between human intragastric environments and cell culture might be a factor for this phenomenon.

Among the CSG and PCD cases, the *H. pylori*-infected group had significantly elevated ENO1 mRNA levels in comparison with the uninfected group. Moreover, the difference in ENO1 between the *H. pylori+* and the *H. pylori-* groups among the CSG cases was much greater than that among the PCD cases or the gastric cancer cases (Table 4). Furthermore, differential ENO1 expression was detected only between the cancer tissues and paracancerous tissues, but not between the *H. pylori*-positive cancer and the *H. pylori*-negative cancer specimens. These findings suggest the effect of *H. pylori* on ENO1 might vary according to periods in the gastric pathogenesis. In the CGS and PCD stages, *H. pylori* may accelerate the pathogenic progression toward the precancerous and cancerous lesions by up-regulating ENO1 expression, and thus increase the risk of gastric cancer. 

Certain reports show that various functions of ENO1 might be related to its subcellular location [21,22,23]. For instance, ENO1 in cytoplasm is involved in glycolysis pathway and aerobic glycolysis of certain tumor cells [21]. In the membrane, ENO1 can promote the invasion and metastasis of carcinoma through activating fibrinolytic enzyme [22]. Localized at the nucleus, ENO1 can inhibit transcription of the original cancer gene c-*myc* [23], and interfering with ENO1 expression can inhibit the growth, migration and invasion of glioma cells [24]. The present study observed that the overexpressed ENO1 was mainly located at the cytoplasm and membrane of the gastric cancer cells, suggesting ENO1 might participate in aerobic glycolysis and promote gastric carcinoma proliferation, invasion and metastasis.

In order to further ascertain the effect of *H. pylori* on ENO1 expression in gastric cells, the present study co-cultured AGS/GES-1 with *H. pylori* MEL-Hp27. The results showed that with the increase of MOI, the ENO1 mRNA levels in the AGS/GES-1 cells also increased. However, a remarkable difference was found between the AGS and GES-1 cells in response to *H. pylori* infection. Co-culture of GES-1 with *H. pylori* showed that the infected cells had significantly reduced ENO1 levels at an MOI of 50:1, similar ENO1 level at 100:1 and significantly increased level at 200:1 when compared with the uninfected cells. As for the AGS cells, no lowered ENO1 level was detected in the infected group at any MOI used, in comparison with the uninfected control. The findings are of significance for understanding certain manifestations of *H. pylori* infection. For instance, most individuals with low infection may carry this bacterium long term without any clinical manifestations, and the occurrence of *H. pylori*-related diseases is related to the bacterial load [25]. The findings evidence that the impact of *H. pylori* infection on the crucial pathogenic factor ENO1 depends to an extent on the bacterial burden. This viewpoint supports that vaccination and antibiotic-based therapies may take effect in the prevention of *H. pylori*-related gastric cancer by only reducing gastric bacterial colonization. This finding is of importance for clinical practices.

Besides the MOIs, the co-culture duration also made a difference in ENO1 expression. At MOI = 200:1, the GES-1 cells had significantly enhanced ENO1 mRNA at 24 h but not at 2 h, 6 h or 12 h of co-culture, while the AGS cells showed significantly decreased ENO1 levels at 2 h and 6 h and increased levels at 12 h and 24 h of co-culture. The findings indicate that not only is the ENO1 expression level related to the co-culture duration, but also that it differs between the cancer cell and the normal cell in ENO1 response to *H. pylori* infection.

The ENO1 expression level in the gastric cells varied from the MOI and infection duration, which might be attributed to the effect of *H. pylori* virulence factors on the gastric cells, e.g., changing the balance of cellular proliferation and apoptosis [26,27]. It might be supposed that during *H. pylori* infection of AGS for 2 h and 6 h, bacterial virulence factors mainly caused apoptosis, and thus lowered the ENO1 expression, while 12 h or 24 h. *pylori* infection predominantly promoted proliferation of the AGS cells and thus up-regulated the ENO1 expression [26,27]. As for the GES-1 cells, differential ENO1 expression was detected in co-culture for 24 h but not 12 h or less, suggesting the cultured cancer cell AGS is more sensitive in ENO1 response to *H. pylori* infection than the normal cell GES-1. 

To further ascertain the mechanism behind the effect of *H. pylori* on ENO1 in the gastric cells, the present study investigated the role of the *cagA* gene in regulation of ENO1 production by cellular transfection. The results showed *cagA* up-regulation of ENO1 expression in both AGS and GES-1 cells. Moreover, the ENO1 mRNA levels were closely correlated with the amount of *cagA* mRNA in the cells. Notably, at 48 h of culture, ENO1 levels in the *cagA*+ GES-1 cells were similar to those in the *cagA*+ AGS cells. However, at 72 h of culture, the *cagA*+ GES-1 cells produced ENO1 more than two times higher than the *cagA*+ AGS cells (Figure 5). These findings evidence that *cagA* is capable of regulating ENO1 expression in both gastric normal cells and cancer cells, but the normal cells produce much higher ENO1 levels than the cancer cells in response to *cagA* transfection. The findings can help understand the high ENO1 expression in the CSG specimens (Table 4), which might be produced by the relatively normal gastric epithelial cells in the CSG tissues under CagA impact.

## 5. Conclusions

This study reveals that ENO1 expression is associated with *H. pylori* (*cagA*+) infection, *cagA* transfection, co-culture duration, MOI, gastric normal/cancer cell lines and cellular differentiation. The findings on the differential regulation of ENO1 expression in gastric normal cells compared with cancer cells in response to *H. pylori* (*cagA*+) infection and *cagA* transfection may be a crucial basis for further ascertaining *H. pylori* pathogenic mechanism and for formulating novel therapeutic and diagnostic strategies.

## Figures and Tables

**Figure 1 medicina-58-01453-f001:**
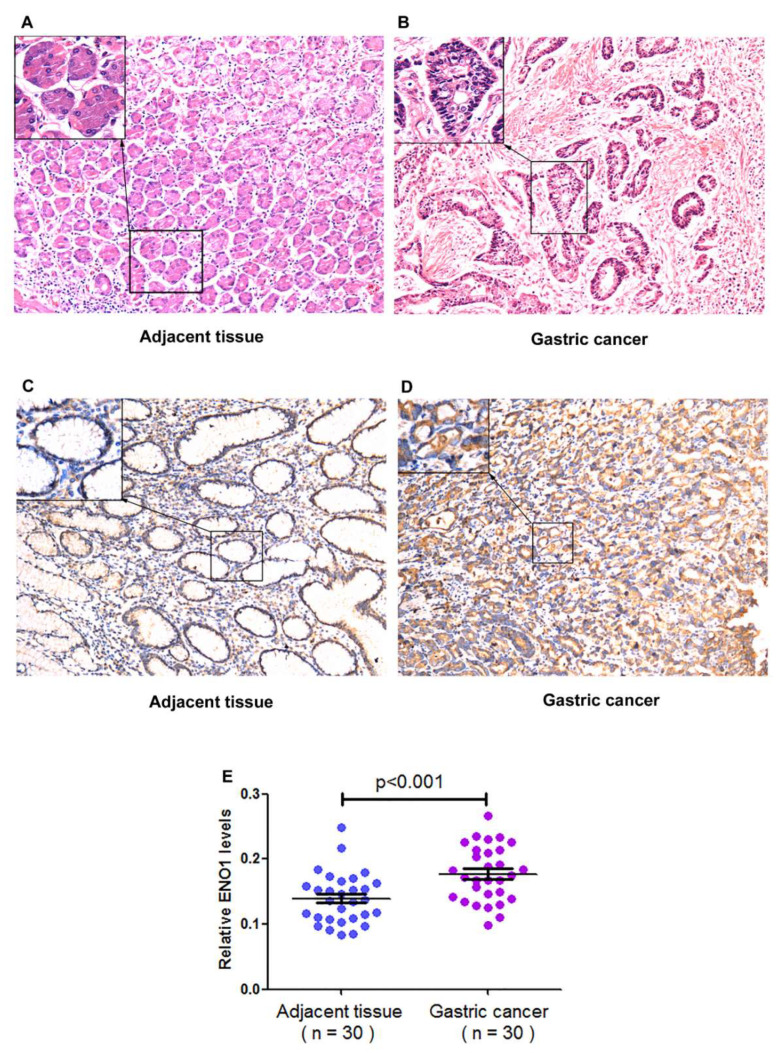
Comparison of gastric cancerous and paracancerous tissues. (**A**,**B**) Gastric adjacent and cancer tissues embedded in paraffin and stained with hematoxylin and eosin (HE). (**C**,**D**) Gastric paracancerous and cancerous tissues embedded in paraffin and stained using the immunohistochemical method. Rabbit anti-ENO1 antibody was used as the primary antibody with a rabbit anti-actin antibody as the control, and radish peroxidase-labeled goat anti-rabbit IgG antibody as the secondary antibody. The positive reaction was visualized using DAB, and the tissue section was re-stained with Harris’ haematoxylin. (**E**) Optical density of the stained gastric cancer sections (n = 30) and paracancerous tissues sections (n = 30) measured using the software ImageJ.

**Figure 2 medicina-58-01453-f002:**
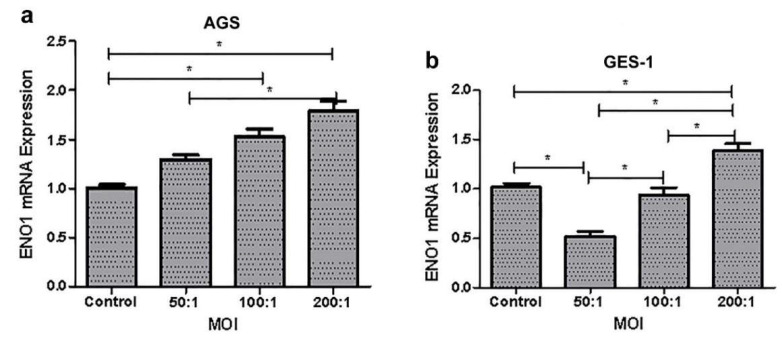
ENO1 mRNA levels in the AGS (**a**) and GES-1 (**b**) cells co-cultured with *H. pylori* for 24 h at a variety of MOIs. On comparing groups in pairs, the statistical standard was adjusted to α´ = 0.008. *, *p* < 0.008.

**Figure 3 medicina-58-01453-f003:**
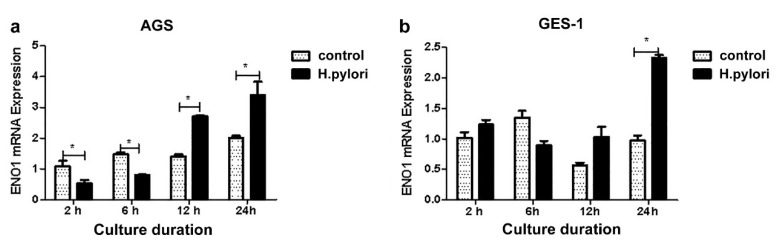
ENO1 mRNA levels in the AGS (**a**) and the GES-1 (**b**) cells co-cultured with *H. pylori* for various times. *, *p* < 0.05.

**Figure 4 medicina-58-01453-f004:**
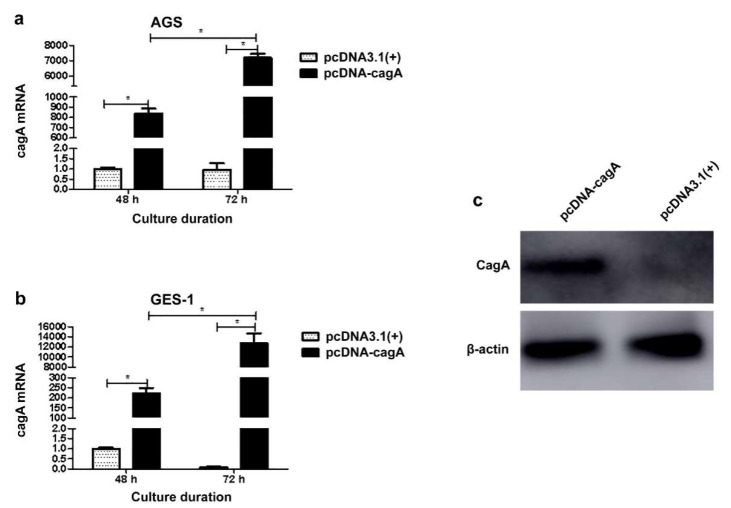
Transfection of the gastric cells and the *cagA* expression. (**a**,**b**) *cagA* mRNA levels in the AGS cells and the GES-1 cells transinfected with pcDNA-*cagA* and pcDNA3.1(+), respectively. (**c**) Expression of the CagA protein in the GES-1 cells transinfected with pcDNA-*cagA* and pcDNA3.1(+), respectively. * *p* < 0.05.

**Figure 5 medicina-58-01453-f005:**
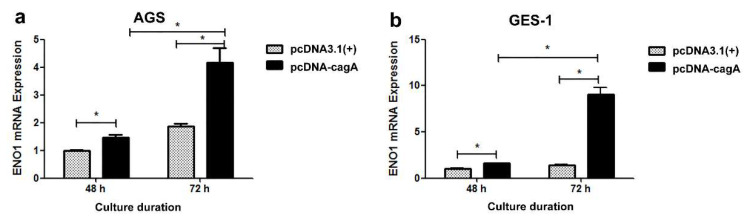
ENO1 mRNA expression in the gastric cells AGS (**a**) and GES-1 (**b**) post transfection with pcDNA-*cagA* and pcDNA3.1(+), respectively. *, *p* < 0.05. At 72 h post transfection, the *cagA+* GES-1 cells produced a much higher ENO1 level than the *cagA*^−^ GES-1 cells and the *cagA*^+^ AGS cells.

**Table 1 medicina-58-01453-t001:** The primers used here for qPCR.

Primer	Length	Sequence	Length of Products
ENO1 sense	19 bp	5′-TGTACCGCCACATCGCTGA-3′	147 bp
ENO1 anti-sense	21 bp	5′-TGAAGTTTGCTGCACCGACTG-3′	
18S rRNA sense	20 bp	5′-CAGCCACCCGAGATTGAGCA-3′	244 bp
18S rRNA anti-sense	20 bp	5′-TAGTAGCGACGGGCGGGTGT-3′	

Note: The procedure of qPCR (7500-fast method) included: Step 1, 95 °C for 30 s; step 2, 95 °C for 5 s, 60 °C for 34 s, 40 cycles; step 3, 95 °C for 15 s, 60 °C for 1 min, 95 °C for 15 s.

**Table 2 medicina-58-01453-t002:** Profiles of the involved patients.

Characters	CSG (n = 73)	PCD (n = 39)	GC (n = 30)	*p*
Age (years)	50.25 ± 13.21	56.69 ± 11.48	60.50 ± 10.03	<0.001
Male	42	21	25	0.023
Female	31	18	5	
*H. pylori* +	42	22	21	0.433
*H. pylori* −	31	17	9	

Note: CSG, chronic superficial gastritis; PCD, precancerous disease; GC, gastric cancer.

**Table 3 medicina-58-01453-t003:** Expression levels of ENO1 in the gastric cancerous tissues.

Characters	Groups	n	ENO1 Protein	*p*
*H. pylori* Infection	*H. pylori* +	21	0.171 ± 0.043	0.244
	*H. pylori* −	9	0.191 ± 0.039	
Age	<55	8	0.188 ± 0.040	0.425
	≥55	22	0.173 ± 0.043	
Location	Cancerous tissues	30	0.177 ± 0.042	<0.001
	Paracancerous tissues	30	0.139 ± 0.039	
Lymphatic metastasis	+	19	0.183 ± 0.044	0.305
	−	11	0.167 ± 0.039	
Differentiation grade	Low	16	0.162 ± 0.040	0.038
	Moderate/High	14	0.194 ± 0.039	

Note: ENO1 expression was enhanced in the cancerous tissues compared with the adjacent tissues (*p* < 0.001). ENO1 levels were significantly higher in the cancerous tissues of moderate/high differentiation than in those of low differentiation (*p* < 0.05).

**Table 4 medicina-58-01453-t004:** Correlation between ENO1 mRNA and *H. pylori* infection in the gastric tissues.

Disease	*H. pylori*	n	ENO1 mRNA	*p*
CSG^§^	+	42	5.93 ± 7.28	0.037
	−	31	3.89 ± 5.01	
PCD^§^	+	22	2.44 ± 2.52	0.630
	−	17	1.66 ± 1.24	
CSG^§^+PCD^§^	+	64	4.37 ± 5.19	0.047
	−	48	2.89 ± 3.37	
GC^Δ^	+	21	0.17 ± 0.04	0.244
	−	9	0.19 ± 0.04	

Note: CSG, chronic superficial gastritis; PCD, precancerous disease; §, fresh specimen; Δ, formalin-fixed specimen. Among the CSG and PCD cases, ENO1 mRNA expression was significantly up-regulated in the *H. pylori*-positive group compared with the negative group (*p <* 0.05). Of the GC cases, no significant ENO1 difference was detected between the *H. pylori+ and H. pylori*- groups.

## Data Availability

Data available on request from the authors.

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
