# Peer review of "Differential Expression of α-Enolase in Clinical Gastric Tissues and Cultured Normal/Cancer Cells in Response to Helicobacter pylori Infection and cagA Transfection"

_medicina, 2022, doi:10.3390/medicina58101453_

Round 1

Reviewer 1 Report

1.       No abbreviations in the text - please complete.

2.       No enlargement on the photo / in the description under Fig 1 - please complete.

3.       How do I select MOI H. pylori doses?

4.       No results presenting cytotoxicity results for H. pylori treated cell lines - please complete

5.       Please standardize the scales (Fig 2). Differently distorting the comparison of the gene expression level for two lines - please correct. what-does the control in chart 2a mean? Please explain the abbreviation.

6.       Fig 2 / Fig3 - was the contemporaneous positive for the level of gene expression? - please explain.

7.       Fig 4/ Fig 5 different scales on the chart - please correct.

8.       Fig 4/Fig 5  what was a control

Author Response

Dear Reviewer,

 As the authors of the manuscript medicina-1892475, we thank you for the constructive comments on our work. Now, we have revised the manuscript carefully as instructed, responded to all the commentspoint by point, prepared the manuscript adhered to the editorial requirements, and submitted the revised version of the manuscript with marked changes. Our responses to your comments are listed below.

Sincerely yours,

 Dr. Rongguang Zhang

Dept. of Epidemiology, College of Public Health

Zhengzhou University, Zhengzhou 450001, China

====================

Reviewer reports:

Reviewer #1: Comments and Suggestions for Authors

  1. No abbreviations in the text - please complete.

Authors: As instructed, we have completed the abbreviations in the text, please see them in the line 354-356, page 11.

  1. No enlargement on the photo / in the description under Fig 1- please complete.

Authors: As instructed, we have added data on enlargement in the description under Fig 1, please see them in page 6.

  1. How do I select MOI H. pylori doses?

Authors: The MOI of H. pylori doses can be selected by carrying out pilot experiments or referring to published papers including ours.

  1. No results presenting cytotoxicity results for H. pylori treated cell lines - please complete

Authors: The cytotoxicity for H. pylori treated cell lines was not included in this study. However, the idea of the reviewer is interesting and may result in significant findings, thus we will consider doing it in future project. Thank you.

  1. Please standardize the scales (Fig 2). Differently distorting the comparison of the gene expression level for two lines - please correct. what-does the control in chart 2a mean? Please explain the abbreviation.

Authors: As instructed, we have standardized the scales (Fig 2) and corrected the mistake. The control in chart 2a mean un-infection. The abbreviation C means Control in original edition, now we use the word Control instead of C in the revised edition (in line 228, page 6).

  1. Fig 2 / Fig3 - was the contemporaneous positive for the level of gene expression? - please explain.

Authors: Yes, Fig 2 / Fig3 shows the contemporaneous positive for the level of gene expression. ENO1 gene is expressed in both infected cells and un-infected cells, although at different levels.

  1. Fig 4/ Fig 5 different scales on the chart - please correct.

Authors: As instructed, we have standardized the scales (Fig 4/ Fig 5) and corrected the mistakes.

  1. Fig 4/Fig 5 what was a control

Authors: In Fig 4/Fig 5, the cells (AGS/Ges-1) transfected with the empty vector pcDNA3.1 (namely without cagA gene insected) were used as controls.

Thank you for your time.

Reviewer 2 Report

All in all it would be an interesting study concerning the expansion of α-enolase in gastric tissues (pathological and normal, including cancer).

Α-enolase is known to be a glycolytic enzyme that is involved in cell proliferation and apoptosis.

The study is well thought out and the research as well as the analyzes are well done.

Only comment, what would be the expression of α-enolase in patients not affected by H.pylori?

Author Response

Dear Reviewer,

As the authors of the manuscript medicina-1892475, we thank you for the constructive comments on our work. Now, we have revised the manuscript carefully, responded to all the commentspoint by point, prepared the manuscript adhered to the editorial requirements, and submitted the revised version of the manuscript with marked changes. Our responses to your comments are listed below.

Sincerely yours,

Dr. Rongguang Zhang

Dept. of Epidemiology, College of Public Health

Zhengzhou University, Zhengzhou 450001, China

==============

Reviewer reports:

Reviewer #2: Comments and Suggestions for Authors

  1. All in all it would be an interesting study concerning the expansion of α-enolase in gastric tissues (pathological and normal, including cancer).

Authors: Yes, thank you.

  1. Α-enolase is known to be a glycolytic enzyme that is involved in cell proliferation and apoptosis.

Authors: Yes, thank you.

  1. The study is well thought out and the research as well as the analyzes are well done.

Authors: Yes, thank you.

  1. Only comment, what would be the expression of α-enolase in patients not affected by pylori?

Authors: ENO1 (α-enolase) is expressed in both H. pylori-infected patients and the patients not affected by H. pylori, although at various levels.

In addition, we have also checked carefully the language used in the manuscript and corrected mistakes. Thank you very much

Round 2

Reviewer 1 Report

The authors responded to your comments. I have no objections.

Author Response

Dear Reviewer,

As the authors of the manuscript medicina-1892475, we thank you for the constructive comments on our work. Now, we have revised the manuscript carefully as instructed, responded to all the comments, point by point, prepared the manuscript adhered to the editorial requirements, and resubmitted the revised version of the manuscript with marked changes. Our responses to your comments are listed below.

Sincerely yours,

Dr. Rongguang Zhang

Dept. of Epidemiology, College of Public Health

Zhengzhou University, Zhengzhou 450001, China

================

Reviewer reports:

Reviewer #1: Comments and Suggestions for Authors

1.English language and style are fine/minor spell check required.

Authors: As instructed, we have checked carefully the language used in the manuscript and corrected mistakes as shown in the revised manuscript.

2. The conclusions can be improved.

Authors: As instructed, we have also checked the conclusions carefully and used a word more accurate here as shown in the revised manuscript.

Thank you so much!
